# Entropic GANs meet VAEs: A Statistical Approach to Compute Sample Likelihoods in GANs

## Abstract

Building on the success of deep learning, two modern approaches to learn a probability model of the observed data are Generative Adversarial Networks (GANs) and Variational AutoEncoders (VAEs). VAEs consider an explicit probability model for the data and compute a generative distribution by maximizing a variational lower-bound on the log-likelihood function. GANs, however, compute a generative model by minimizing a distance between observed and generated probability distributions without considering an explicit model for the observed data. The lack of having explicit probability models in GANs prohibits computation of sample likelihoods in their frameworks and limits their use in statistical inference problems. In this work, we show that an optimal transport GAN with the entropy regularization can be viewed as a generative model that maximizes a lower-bound on average sample likelihoods, an approach that VAEs are based on. In particular, our proof constructs an explicit probability model for GANs that can be used to compute likelihood statistics within GAN's framework. Our numerical results on several datasets demonstrate consistent trends with the proposed theory.

## 1 Introduction

Learning generative models is becoming an increasingly important problem in machine learning and statistics with a wide range of applications in self-driving cars (Santana & Hotz, 2016), robotics (Hirose et al., 2017), natural language processing (Lee & Tsao, 2018), domain-transfer (Sankaranarayanan et al., 2018), computational biology (Ghahramani et al., 2018), etc. Two modern approaches to deal with this problem are Generative Adversarial Networks (GANs) (Goodfellow et al., 2014) and Variational AutoEncoders (VAEs) (Kingma & Welling, 2013; Makhzani et al., 2015; Rosca et al., 2017; Tolstikhin et al., 2017; Mescheder et al., 2017b).

VAEs (Kingma & Welling, 2013) compute a generative model by maximizing a variational lower-bound on average sample likelihoods using an explicit probability distribution for the data. GANs, however, learn a generative model by minimizing a distance between observed and generated distributions without considering an explicit probability model for the data. Empirically, GANs have been shown to produce higher-quality generative samples than that of VAEs (Karras et al., 2017). However, since GANs do not consider an explicit probability model for the data, we are unable to compute sample likelihoods using their generative models. Computations of sample likelihoods and *posterior* distributions of latent variables are critical in several statistical inference. Inability to obtain such statistics within GAN's framework severely limits their applications in such statistical inference problems.

In this paper, we resolve these issues for a general formulation of GANs by providing a theoretically-justified approach to compute sample likelihoods using GAN's generative model. Our results can open new directions to use GANs in massive-data applications such as model selection, sample selection, hypothesis-testing, etc (see more details in Section 5).

Now, we state our main results *informally* without going into technical conditions while precise statements of our results are presented in Section 2. Let $Y$ and $\hat{Y} := \mathbf{G}(X)$ represent observed (i.e. real) and generative (i.e. fake or synthetic) variables, respectively. $X$ (i.e. the latent variable) is the randomness used as the input to the generator $\mathbf{G}(.)$. Consider the following explicit probability

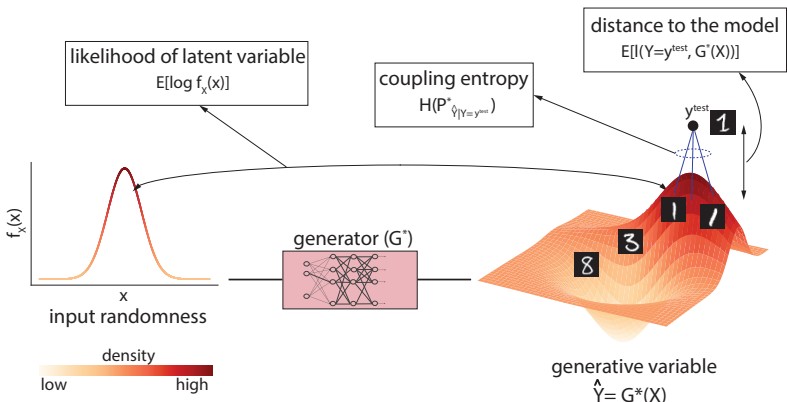

Figure 1: A statistical framework for GANs. By training a GAN model, we first compute optimal generator $\mathbf{G}^*$ and optimal coupling between the observed variable $Y$ and the latent variable $X$. The likelihood of a test sample $\mathbf{y}^{\text{test}}$ can then be lower-bounded using a combination of three terms: (1) the expected distance of $\mathbf{y}^{\text{test}}$ to the distribution learnt by the generative model, (2) the entropy of the coupled latent variable given $\mathbf{y}^{\text{test}}$ and (3) the likelihood of the coupled latent variable with $\mathbf{y}^{\text{test}}$.

model of the data given a latent sample $X = \mathbf{x}$:

$$f_{Y|X=\mathbf{x}}(\mathbf{y}) \propto \exp(-\ell(\mathbf{y}, \mathbf{G}(\mathbf{x}))), \tag{1.1}$$

where $\ell(.,.)$ is a loss function. $f_{Y|X=\mathbf{x}}(\mathbf{y})$ is the model that we are considering for the underlying data distribution. This is a reasonable model for the data as the function $\mathbf{G}$ can be a complex function. Similar data models have been used in VAEs. Under this explicit probability model, we show that minimizing the objective of an *optimal transport* GAN (e.g. Wasserstein GAN Arjovsky et al. (2017)) with the cost function $\ell(.,.)$ and an entropy regularization (Cuturi, 2013; Seguy et al., 2017) maximizes a variational lower-bound on average sample likelihoods. I.e.

$$\text{average sample likelihoods} \geq -(\text{entropic GAN objective}) + \text{constants}. \tag{1.2}$$

If $\ell(\mathbf{y}, \hat{\mathbf{y}}) = \|\mathbf{y} - \hat{\mathbf{y}}\|_2$, the optimal transport (OT) GAN simplifies to WGAN (Arjovsky et al., 2017) while if $\ell(\mathbf{y}, \hat{\mathbf{y}}) = \|\mathbf{y} - \hat{\mathbf{y}}\|_2^2$, the OT GAN simplifies to the quadratic GAN (or, W2GAN) (Feizi et al., 2017). The precise statement of this result can be found in Theorem 1. This result provides a statistical justification for GAN's optimization and puts it in par with VAEs whose goal is to maximize a lower bound on sample likelihoods. We note that the entropy regularization has been proposed primarily to improve computational aspects of GANs (Genevay et al., 2018). Our results provide an additional statistical justification for this regularization term. Moreover, using GAN's training, we obtain a coupling between the observed variable $Y$ and the latent variable $X$. This coupling provides the conditional distribution of the latent variable $X$ given an observed sample $Y = \mathbf{y}$. The explicit model of equation 1.1 acts similar to the *decoder* in the VAE framework, while the coupling computed using GANs acts as an *encoder*.

Connections between GANs and VAEs have been investigated in some of the recent works as well (Hu et al., 2018; Mescheder et al., 2017a). In Hu et al. (2018), GANs are interpreted as methods performing variational inference on a generative model in the label space. In their framework, observed data samples are treated as latent variables while the generative variable is the indicator of whether data is real or fake. The method in Mescheder et al. (2017a), on the other hand, uses an auxiliary discriminator network to rephrase the maximum-likelihood objective of a VAE as a two-player game similar to the objective of a GAN. Our method is different from both these approaches as we consider an explicit probability model for the data, and show that the entropic GAN objective maximizes a variational lower bound under this probability model, thus allowing sample likelihood computation in GANs similar to VAEs.

Of relevance to our work is Wu et al. (2016), in which annealed importance sampling (AIS) is used to evaluate the approximate likelihood of decoder-based generative models. More specifically, a Gaussian observation model with a fixed variance is used as the generative distribution for GAN-based models on which the AIS is computed. Gaussian observation models may not be proper

specially in high-dimensional spaces. Our approach, on the other hand, makes a connection between GANs and VAEs by constructing a theoretically-motivated model for the data distribution in GANs. We then leverage this approach in computing sample likelihood estimates in GANs.

Another key question that we address here is how to estimate the likelihood of a new sample $\mathbf{y}^{\text{test}}$ given the generative model trained using GANs. For instance, if we train a GAN on *stop-sign* images, upon receiving a new image, one may wish to compute the likelihood of the new sample $\mathbf{y}^{\text{test}}$ according to the trained generative model. In standard GAN formulations, the support of the generative distribution lies on the range of the optimal generator function. Thus, if the observed sample $\mathbf{y}^{\text{test}}$ does not lie on that range (which is very likely in practice), there is no way to assign a sensible likelihood score to that sample. Below, we show that using the explicit probability model of equation 1.1, we can *lower-bound* the likelihood of this sample $\mathbf{y}^{\text{test}}$. This is similar to the variational lower-bound on sample likelihoods used in VAEs. Our numerical results show that this lower-bound well-reflect the expected trends of the true sample likelihoods.

Let $\mathbf{G}^*$ and $\mathbb{P}^*_{Y,X}$ be the optimal generator and the optimal coupling between real and latent variables, respectively. The optimal coupling $\mathbb{P}^*_{Y,X}$ can be computed efficiently for entropic GANs as we explain in Section 3. For other GAN architectures, one may approximate such couplings as we explain in Section 4. The log likelihood of a new test sample $\mathbf{y}^{\text{test}}$ can be lower-bounded as

$$\underbrace{\log f_Y(\mathbf{y}^{\text{test}})}_{\text{log likelihood}} \geq \underbrace{-\mathbb{E}_{\mathbb{P}^*_{X|Y=\mathbf{y}^{\text{test}}}}\left[\ell(\mathbf{y}^{\text{test}}, \mathbf{G}^*(\mathbf{x}))\right]}_{\text{distance to the generative model}} + \underbrace{H\left(\mathbb{P}^*_{X|Y=\mathbf{y}^{\text{test}}}\right)}_{\text{coupling entropy}} + \underbrace{\mathbb{E}_{\mathbb{P}^*_{X|Y=\mathbf{y}^{\text{test}}}}\left[-\frac{\|\mathbf{x}\|^2}{2}\right]}_{\text{likelihood of latent variable}}. \qquad (1.3)$$

We present the precise statement of this result in Corollary 2. This result combines three components in order to approximate the likelihood of a sample given a trained generative model:

- The distance between $\mathbf{y}^{\text{test}}$ to the generative model. If this distance is large, the likelihood of observing $\mathbf{y}^{\text{test}}$ from the generative model is small.
- The entropy of the coupled latent variable. If the entropy term is large, the coupled latent variable has a large randomness. This contributes positively to the sample likelihood.
- The likelihood of the coupled latent variable. If latent samples have large likelihoods, the likelihood of the observed test sample will be large as well.

Figure 2a provides a pictorial illustration of these components. In what follows, we explain the technical ingredients of our main results. In Section 3, we present computational methods for GANs and entropic GANs, while in Section 4, we provide numerical experiments on benchmark datasets.

## 2 MAIN RESULTS

Let $Y \in \mathbb{R}^d$ represent the real-data random variable with a probability density function $f_Y(\mathbf{y})$. GAN's goal is to find a generator function $\mathbf{G} : \mathbb{R}^r \to \mathbb{R}^d$ such that $\hat{Y} := \mathbf{G}(X)$ has a similar distribution to $Y$. Let $X$ be an $r$-dimensional random variable with a fixed probability density function $f_X(\mathbf{x})$. Here, we assume $f_X(.)$ is the density of a normal distribution. In practice, we observe $m$ samples $\{\mathbf{y}_1, ..., \mathbf{y}_m\}$ from $Y$ and generate $m'$ samples from $\hat{Y}$, i.e., $\{\hat{\mathbf{y}}_1, ..., \hat{\mathbf{y}}_{m'}\}$ where $\hat{\mathbf{y}}_i = \mathbf{G}(\mathbf{x}_i)$ for $1 \leq i \leq m$. We represent these empirical distributions by $\mathbb{P}_Y$ and $\mathbb{P}_{\hat{Y}}$, respectively. Note that the number of generative samples $m'$ can be arbitrarily large.

GAN computes the optimal generator $\mathbf{G}^*$ by minimizing a distance between the observed distribution $\mathbb{P}_Y$ and the generative one $\mathbb{P}_{\hat{Y}}$. Common distance measures include *optimal transport* measures (e.g. Wasserstein GAN (Arjovsky et al., 2017), WGAN+Gradient Penalty (Gulrajani et al., 2017), GAN+Spectral Normalization (Miyato et al., 2018), WGAN+Truncated Gradient Penalty (Petzka et al., 2017), relaxed WGAN (Guo et al., 2017)), and *divergence* measures (e.g. the original GAN's formulation (Goodfellow et al., 2014), $f$-GAN (Nowozin et al., 2016)), etc.

In this paper, we focus on GANs based on optimal transport (OT) distance (Villani, 2008; Arjovsky et al., 2017) defined for a general loss function $\ell(.,.)$ as follows

$$W_\ell(\mathbb{P}_Y, \mathbb{P}_{\hat{Y}}) := \min_{\mathbb{P}_{Y,\hat{Y}}} \mathbb{E}\left[\ell(Y, \hat{Y})\right]. \qquad (2.1)$$

$\mathbb{P}_{Y,\hat{Y}}$ is the joint distribution whose marginal distributions are equal to $\mathbb{P}_Y$ and $\mathbb{P}_{\hat{Y}}$, respectively. If $\ell(\mathbf{y}, \hat{\mathbf{y}}) = \|\mathbf{y} - \hat{\mathbf{y}}\|_2$, this distance is called the first-order Wasserstein distance and is referred to by $W_1(.,.)$, while if $\ell(\mathbf{y}, \hat{\mathbf{y}}) = \|\mathbf{y} - \hat{\mathbf{y}}\|_2^2$, this measure is referred to by $W_2(.,.)$ where $W_2$ is the second-order Wasserstein distance (Villani, 2008).

The optimal transport (OT) GAN is formulated using the following optimization (Arjovsky et al., 2017; Villani, 2008):

$$\min_{\mathbf{G} \in \mathcal{G}} W_\ell(\mathbb{P}_Y, \mathbb{P}_{\hat{Y}}), \tag{2.2}$$

where $\mathcal{G}$ is the set of generator functions. Examples of the OT GAN are WGAN (Arjovsky et al., 2017) corresponding to the first-order Wasserstein distance $W_1(.,.)$ [1] and the quadratic GAN (or, the W2GAN) (Feizi et al., 2017) corresponding to the second-order Wasserstein distance $W_2(.,.)$.

Note that optimization 2.2 is a *min-min* optimization. The objective of this optimization is not smooth in $\mathbf{G}$ and it is often computationally expensive to obtain a solution (Sanjabi et al., 2018). One approach to improve computational aspects of this optimization is to add a regularization term to make its objective *strongly* convex (Cuturi, 2013; Seguy et al., 2017). The Shannon entropy function is defined as $H(\mathbb{P}_{Y,\hat{Y}}) := -\mathbb{E}\left[\log \mathbb{P}_{Y,\hat{Y}}\right]$. The negative Shannon entropy is a common strongly-convex regularization term. This leads to the following optimal transport GAN formulation with the entropy regularization, or for simplicity, the *entropic GAN* formulation:

$$\min_{\mathbf{G} \in \mathcal{G}} \min_{\mathbb{P}_{Y,\hat{Y}}} \mathbb{E}\left[\ell(Y, \hat{Y})\right] - \lambda H\left(\mathbb{P}_{Y,\hat{Y}}\right), \tag{2.3}$$

where $\lambda$ is the regularization parameter.

There are two approaches to solve the optimization problem 2.3. The first approach uses an iterative method to solve the *min-min* formulation (Genevay et al., 2017). Another approach is to solve an equivelent *min-max* formulation by writing the dual of the inner minimization (Seguy et al., 2017; Sanjabi et al., 2018). The latter is often referred to as a GAN formulation since the min-max optimization is over a set of generator functions and a set of discriminator functions. The details of this approach are further explained in Section 3.

In the following, we present an explicit probability model for entropic GANs under which their objective can be viewed as maximizing a lower bound on average sample likelihoods.

**Theorem 1** *Let the loss function be shift invariant, i.e., $\ell(\mathbf{y}, \hat{\mathbf{y}}) = h(\mathbf{y} - \hat{\mathbf{y}})$. Let*

$$f_{Y|X=\mathbf{x}}(\mathbf{y}) = C \exp(-\ell(\mathbf{y}, \mathbf{G}(\mathbf{x}))/\lambda), \tag{2.4}$$

*be an explicit probability model for $Y$ given $X = \mathbf{x}$ for a well-defined normalization*

$$C := \frac{1}{\int_{\mathbf{y} \in \mathbb{R}^d} \exp(-\ell(\mathbf{y}, G(\mathbf{x}))/\lambda)}. \tag{2.5}$$

*Then, we have*

$$\underbrace{\mathbb{E}_{\mathbb{P}_Y}\left[\log f_Y(Y)\right]}_{\text{ave. sample likelihoods}} \geq -\frac{1}{\lambda} \underbrace{\left\{\mathbb{E}_{\mathbb{P}_{Y,\hat{Y}}}\left[\ell(Y, \hat{Y})\right] - \lambda H\left(\mathbb{P}_{Y,\hat{Y}}\right)\right\}}_{\text{entropic GAN objective}} + constants. \tag{2.6}$$

*In words, the entropic GAN maximizes a lower bound on sample likelihoods according to the explicit probability model of equation 2.4.*

The proof of this theorem is presented in Section A. This result has a similar flavor to that of VAEs (Makhzani et al., 2015; Rosca et al., 2017; Tolstikhin et al., 2017; Mescheder et al., 2017b) where a generative model is computed by maximizing a lower bound on sample likelihoods.

Having a shift invariant loss function is critical for Theorem 1 as this makes the normalization term $C$ independent from $\mathbf{G}$ and $\mathbf{x}$ (to see this, one can define $\mathbf{y}' := \mathbf{y} - \mathbf{G}(\mathbf{x})$ in equation 2.6). The most

---

[1] Note some references (e.g. (Arjovsky et al., 2017)) refer to the first-order Wasserstein distance simply as the Wasserstein distance. In this paper, we distinguish between different Wasserstein distances explicitly.

standard OT GAN loss functions such as the $L_2$ for WGAN (Arjovsky et al., 2017) and the quadratic loss for W2GAN (Feizi et al., 2017) satisfy this property.

One can further simplify this result by considering specific loss functions. For example, we have the following result for the entropic GAN with the quadratic loss function.

**Corollary 1** *Let $\ell(\mathbf{y}, \hat{\mathbf{y}}) = \|\mathbf{y} - \hat{\mathbf{y}}\|^2/2$. Then, $f_{Y|X=\mathbf{x}}(.)$ of equation 2.4 corresponds to the multivariate Gaussian density function and $C = \frac{1}{\sqrt{(2\pi\lambda)^d}}$. In this case, the constant term in equation 2.6 is equal to $-\log(m) - d\log(2\pi\lambda)/2 - r/2 - \log(2\pi)/2$.*

Let $\mathbf{G}^*$ and $\mathbb{P}^*_{Y,X}$ be optimal solutions of an entropic GAN optimization 2.3 (note that the optimal coupling can be computed efficiently using equation 3.7). Let $\mathbf{y}^{\text{test}}$ be a newly observed sample. An important question is what the likelihood of this sample is given the trained generative model. Using the explicit probability model of equation 2.4 and the result of Theorem 1, we can (approximately) compute sample likelihoods as explained in the following corollary.

**Corollary 2** *Let $\mathbf{G}^*$ and $\mathbb{P}^*_{Y,\hat{Y}}$ (or, alternatively $\mathbb{P}^*_{Y,X}$) be optimal solutions of the entropic GAN equation 2.3. Let $\mathbf{y}^{test}$ be a new observed sample. We have*

$$\log f_Y(\mathbf{y}^{test}) \geq -\frac{1}{\lambda}\left\{\mathbb{E}_{\mathbb{P}^*_{X|Y=\mathbf{y}^{test}}}\left[\ell(\mathbf{y}^{test}, \mathbf{G}^*(\mathbf{x}))\right] - \lambda H\left(\mathbb{P}^*_{X|Y=\mathbf{y}^{test}}\right)\right\} \quad (2.7)$$

$$+ \mathbb{E}_{\mathbb{P}^*_{X|Y=\mathbf{y}^{test}}}\left[-\frac{\|\mathbf{x}\|^2}{2}\right] + constants.$$

*The inequality becomes tight iff $D_{\mathrm{KL}}\left(\mathbb{P}^*_{X|Y=\mathbf{y}^{test}} \| f_{X|Y=\mathbf{y}^{test}}\right) = 0$ where $D_{\mathrm{KL}}(.\|.)$ is the Kullback-Leibler divergence between two distributions.*

## 3 GAN's Dual Formulation

In this section, we discuss dual formulations for OT GAN (equation 2.2) and entropic GAN (equation 2.3) optimizations. These dual formulations are *min-max* optimizations over two function classes, namely the generator and the discriminator. Often local search methods such as alternating gradient descent (GD) are used to compute a solution for these min-max optimizations.

First, we discuss the dual formulation of OT GAN optimization 2.2. Using the duality of the inner minimization, which is a linear program, we can re-write optimization 2.2 as follows (Villani, 2008):

$$\min_{\mathbf{G} \in \mathcal{G}} \max_{\mathbf{D}_1, \mathbf{D}_2} \mathbb{E}\left[\mathbf{D}_1(Y)\right] - \mathbb{E}\left[\mathbf{D}_2(\mathbf{G}(X))\right], \quad (3.1)$$

where $\mathbf{D}_1(\mathbf{y}) - \mathbf{D}_2(\hat{\mathbf{y}}) \leq \ell(\mathbf{y}, \hat{\mathbf{y}})$ for all $(\mathbf{y}, \hat{\mathbf{y}})$. The maximization is over two sets of functions $\mathbf{D}_1$ and $\mathbf{D}_2$ which are coupled using the loss function. Using the Kantorovich duality Villani (2008), we can further simplify this optimization as follows:

$$\min_{\mathbf{G} \in \mathcal{G}} \max_{\mathbf{D}:\ell-\text{convex}} \mathbb{E}\left[\mathbf{D}(Y)\right] - \mathbb{E}\left[\mathbf{D}^{(\ell)}(\mathbf{G}(X))\right], \quad (3.2)$$

where $\mathbf{D}^{(\ell)}(\hat{Y}) := \inf_Y \ell(Y, \hat{Y}) + \mathbf{D}(Y)$ is the $\ell$-conjugate function of $\mathbf{D}(.)$ and $\mathbf{D}$ is restricted to $\ell$-convex functions (Villani, 2008). The above optimization provides a general formulation for OT GANs. If the loss function is $\|.\|_2$, then the optimal transport distance is referred to as the first order Wasserstein distance. In this case, the min-max optimization 3.2 simplifies to the following optimization (Arjovsky et al., 2017):

$$\min_{\mathbf{G} \in \mathcal{G}} \max_{\mathbf{D}:1\text{-Lip}} \mathbb{E}\left[\mathbf{D}(Y)\right] - \mathbb{E}\left[\mathbf{D}(\mathbf{G}(X))\right]. \quad (3.3)$$

This is often referred to as Wasserstein GAN, or WGAN (Arjovsky et al., 2017). If the loss function is quadratic, then the OT GAN is referred to as the quadratic GAN (or, W2GAN) (Feizi et al., 2017).

Similarly, the dual formulation of the entropic GAN equation 2.3 can be written as the following optimization (Cuturi, 2013; Seguy et al., 2017) [2]:

---

[2]Note that optimization 3.4 is dual of optimization 2.3 when the terms $\lambda H(\mathbb{P}_Y) + \lambda H(\mathbb{P}_{\hat{Y}})$ have been added to its objective. Since for a fixed $\mathbf{G}$ (fixed marginals), these terms are constants, they can be ignored from the optimization objective without loss of generality.

$$\min_{\mathbf{G} \in \mathcal{G}} \max_{\mathbf{D}_1, \mathbf{D}_2} \mathbb{E}\left[\mathbf{D}_1(Y)\right] - \mathbb{E}\left[\mathbf{D}_2(\mathbf{G}(X))\right] - \lambda \mathbb{E}_{\mathbb{P}_Y \times \mathbb{P}_{\hat{Y}}} \left[\exp\left(v(\mathbf{y}, \hat{\mathbf{y}})/\lambda\right)\right], \qquad (3.4)$$

where

$$v(\mathbf{y}, \hat{\mathbf{y}}) := \mathbf{D}_1(\mathbf{y}) - \mathbf{D}_2(\hat{\mathbf{y}}) - \ell(\mathbf{y}, \hat{\mathbf{y}}). \qquad (3.5)$$

Note that the hard constraint of optimization 3.1 is being replaced by a soft constraint in optimization 3.2. In this case, optimal primal variables $\mathbb{P}^*_{Y, \hat{Y}}$ can be computed according to the following lemma (Seguy et al., 2017):

**Lemma 1** *Let $\mathbf{D}_1^*$ and $\mathbf{D}_2^*$ be the optimal discriminator functions for a given generator function $\mathbf{G}$ according to optimization 3.4. Let*

$$v^*(\mathbf{y}, \hat{\mathbf{y}}) := \mathbf{D}_1^*(\mathbf{y}) - \mathbf{D}_2^*(\hat{\mathbf{y}}) - \ell(\mathbf{y}, \hat{\mathbf{y}}). \qquad (3.6)$$

*Then,*

$$\mathbb{P}^*_{Y, \hat{Y}}(\mathbf{y}, \hat{\mathbf{y}}) = \mathbb{P}_Y(\mathbf{y}) \mathbb{P}_{\hat{Y}}(\hat{\mathbf{y}}) \exp\left(v^*(\mathbf{y}, \hat{\mathbf{y}})/\lambda\right). \qquad (3.7)$$

This lemma is important for our results since it provides an efficient way to compute the optimal coupling between real and generative variables (i.e. $P^*_{Y, \hat{Y}}$) using the optimal generator ($\mathbf{G}^*$) and discriminators ($\mathbf{D}_1^*$ and $\mathbf{D}_2^*$) of optimization 3.4. It is worth noting that without the entropy regularization term, computing the optimal coupling using the optimal generator and discriminator functions is not straightforward in general (unless in some special cases such as W2GAN (Villani, 2008; Feizi et al., 2017)). This is another additional computational benefit of using entropic GAN.

## 4 EXPERIMENTAL RESULTS

In this section, we supplement our theoretical results with experimental validations. One of the main objectives of our work is to provide a framework to compute sample likelihoods in GANs. Such likelihood statistics can then be used in several statistical inference applications that we discuss in Section 5. With a trained entropic WGAN, the likelihood of a test sample can be lower-bounded using Corollary 2. Note that this likelihood estimate requires the discriminators $\mathbf{D}_1$ and $\mathbf{D}_2$ to be solved to optimality. In our implementation, we use the algorithm presented in Sanjabi et al. (2018) to train the Entropic GAN. It has been proven (Sanjabi et al., 2018) that this algorithm leads to a good approximation of stationary solutions of Entropic GAN.

To obtain the surrogate likelihood estimates using Corollary 2, we need to compute the density $\mathbb{P}^*_{X|Y=\mathbf{y}^{\text{test}}}(\mathbf{x})$. As shown in Lemma 1, WGAN with entropy regularization provides a closed-form solution to the conditional density of the latent variable (equation 3.7). When $\mathbf{G}^*$ is injective, $\mathbb{P}^*_{X|Y=\mathbf{y}^{\text{test}}}(\mathbf{x})$ can be obtained from equation 3.7 by change of variables. In general case, $\mathbb{P}^*_{X|Y=\mathbf{y}^{\text{test}}}(\mathbf{x})$ is not well defined as multiple $\mathbf{x}$ can produce the same $\mathbf{y}^{\text{test}}$. In this case,

$$\mathbb{P}^*_{\hat{Y}|Y=\mathbf{y}^{\text{test}}}(\hat{\mathbf{y}}) = \sum_{\mathbf{x}|\mathbf{G}^*(\mathbf{x})=\mathbf{y}} \mathbb{P}^*_{X|Y=\mathbf{y}^{\text{test}}}(\mathbf{x}). \qquad (4.1)$$

Also, from equation 3.7, we have

$$\mathbb{P}^*_{\hat{Y}|Y=\mathbf{y}^{\text{test}}}(\hat{\mathbf{y}}) = \sum_{\mathbf{x}|\mathbf{G}^*(\mathbf{x})=\mathbf{y}} \mathbb{P}_X(\mathbf{x}) \exp\left(v^*\left(\mathbf{y}^{\text{test}}, \mathbf{G}^*(\mathbf{x})\right)/\lambda\right). \qquad (4.2)$$

One solution (which may not be unique) that satisfies both equation 4.1 and 4.2 is

$$\mathbb{P}^*_{X|Y=\mathbf{y}^{\text{test}}}(\mathbf{x}) = \mathbb{P}_X(\mathbf{x}) \exp\left(v^*\left(\mathbf{y}^{\text{test}}, G^*(\mathbf{x})\right)/\lambda\right). \qquad (4.3)$$

Ideally, we would like to choose $\mathbb{P}^*_{X|Y=\mathbf{y}^{\text{test}}}(\mathbf{x})$ satisfying equation 4.1 and 4.2 that maximizes the lower bound of Corollary 2. But finding such a solution can be difficult in general. Instead we use equation 4.3 to evaluate the surrogate likelihoods of Corollary 2 (note that our results still hold in this case). In order to compute our proposed surrogate likelihood, we need to draw samples from the distribution $\mathbb{P}^*_{X|Y=\mathbf{y}^{\text{test}}}(\mathbf{x})$. One approach is to use a Markov chain Monte Carlo (MCMC) method to sample from this distribution. In our experiments, however, we found that MCMC demonstrates poor performance owing to the high dimensional nature of $X$. A similar issue with MCMC has been reported for VAEs in Kingma & Welling (2013). Thus, we use a different estimator to compute the likelihood surrogate which provides a better exploration of the latent space. We present our sampling procedure in Alg. 1 of Appendix.

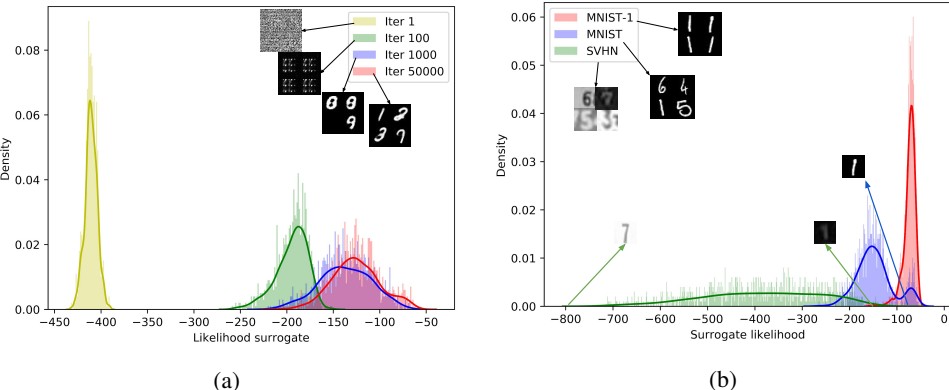

(a)                 (b)

Figure 2: (a) Distributions of surrogate sample likelihoods at different iterations of entropic WGAN's training using MNIST dataset. (b) Distributions of surrogate sample likelihoods of MNIST, MNIST-1 and SVHM datasets using a GAN trained on MNIST-1.

## 4.1 LIKELIHOOD EVOLUTION IN GAN'S TRAINING

In the experiments of this section, we study how sample likelihoods vary during GAN's training. An entropic WGAN is first trained on MNIST dataset. Then, we randomly choose $1,000$ samples from MNIST test-set to compute the surrogate likelihoods using Algorithm 1 at different training iterations. Surrogate likelihood computation requires solving $\mathbf{D}_1$ and $\mathbf{D}_2$ to optimality for a given $\mathbf{G}$ (refer to Lemma. 2), which might not be satisfied at the intermediate iterations of the training process. Therefore, before computing the surrogate likelihoods, discriminators $\mathbf{D}_1$ and $\mathbf{D}_2$ are updated for $100$ steps for a fixed $\mathbf{G}$. We expect sample likelihoods to increase over training iterations as the quality of the generative model improves.

Fig. 2a demonstrates the evolution of sample likelihood distributions at different training iterations of the entropic WGAN. At iteration 1, surrogate likelihood values are very low as GAN's generated images are merely random noise. The likelihood distribution shifts towards high values during the training and saturates beyond a point. Details of this experiment are presented in Appendix E.

## 4.2 LIKELIHOOD COMPARISON ACROSS DIFFERENT DATASETS

In this section, we perform experiments across different datasets. An entropic WGAN is first trained on a subset of samples from the MNIST dataset containing digit 1 (which we call the MNIST-1 dataset). With this trained model, likelihood estimates are computed for (1) samples from the entire MNIST dataset, and (2) samples from the Street View House Numbers (SVHN) dataset (Netzer et al., 2011) (Fig. 2b). In each experiment, the likelihood estimates are computed for 1000 samples. We note that highest likelihood estimates are obtained for samples from MNIST-1 dataset, the same dataset on which the GAN was trained. The likelihood distribution for the MNIST dataset is bimodal with one mode peaking inline with the MNIST-1 mode. Samples from this mode correspond to digit 1 in the MNIST dataset. The other mode, which is the dominant one, contains the rest of the digits and has relatively low likelihood estimates. The SVHN dataset, on the other hand, has much smaller likelihoods as its distribution is significantly different than that of MNIST. Furthermore, we observe that the likelihood distribution of SVHN samples has a large spread (variance). This is because samples of the SVHN dataset is more diverse with varying backgrounds and styles than samples from MNIST. We note that SVHN samples with high likelihood estimates correspond to images that are similar to MNIST digits, while samples with low scores are different than MNIST samples. Details of this experiment are presented in Appendix E.

## 4.3 APPROXIMATE LIKELIHOOD COMPUTATION IN UN-REGULARIZED GANS

Most standard GAN architectures do not have the entropy regularization. Likelihood lower bounds of Theorem 1 and Corollary 2 hold even for those GANs as long as we obtain the optimal coupling $\mathbb{P}^*_{Y,\hat{Y}}$ in addition to the optimal generator $\mathbf{G}^*$ from GAN's training. Computation of optimal cou-

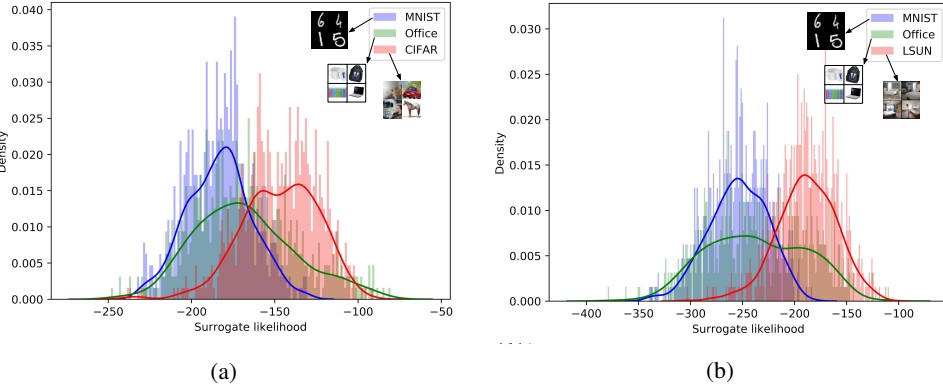

Figure 3: (a) Sample likelihood estimates of MNIST, Office and CIFAR datasets using a GAN trained on the CIFAR dataset. (b) Sample likelihood estimates of MNIST, Office and LSUN datasets using a GAN trained on the LSUN dataset.

pling $\mathbb{P}^*_{Y,\hat{Y}}$ from the dual formulation of OT GAN can be done when the loss function is quadratic (Feizi et al., 2017). In this case, the gradient of the optimal discriminator provides the optimal coupling between $Y$ and $\hat{Y}$ (Villani, 2008) (see Lemma 2 in Appendix C).

For a general GAN architecture, however, the exact computation of optimal coupling $\mathbb{P}^*_{Y,\hat{Y}}$ may be difficult. One sensible approximation is to couple $Y = \mathbf{y}^{\text{test}}$ with a single latent sample $\tilde{\mathbf{x}}$ (we are assuming the conditional distribution $\mathbb{P}^*_{X|Y=\mathbf{y}^{\text{test}}}$ is an impulse function). To compute $\tilde{\mathbf{x}}$ corresponding to a $\mathbf{y}^{\text{test}}$, we sample $k$ latent samples $\{\mathbf{x}'_i\}^k_{i=1}$ and select the $\mathbf{x}'_i$ whose $\mathbf{G}^*(\mathbf{x}'_i)$ is closest to $\mathbf{y}^{\text{test}}$. This heuristic takes into account both the likelihood of the latent variable as well as the distance between $\mathbf{y}^{\text{test}}$ and the model (similarly to equation 3.7). We can then use Corollary 2 to approximate sample likelihoods for various GAN architectures.

We use this approach to compute likelihood estimates for CIFAR-10 (Krizhevsky, 2009) and LSUN-Bedrooms (Yu et al., 2015) datasets. For CIFAR-10, we train DCGAN while for LSUN, we train WGAN (details of these experiments can be found in Appendix E). Fig. 3a demonstrates sample likelihood estimates of different datasets using a GAN trained on CIFAR-10. Likelihoods assigned to samples from MNIST and Office datasets are lower than that of the CIFAR dataset. Samples from the Office dataset, however, are assigned to higher likelihood values than MNIST samples. We note that the Office dataset is indeed more similar to the CIFAR dataset than MNIST. A similar experiment has been repeated for LSUN-Bedrooms (Yu et al., 2015) dataset. We observe similar performance trends in this experiment (Fig. 3b).

## 5  CONCLUSION

In this paper, we have provided a statistical framework for a family of GANs. Our main result shows that the entropic GAN optimization can be viewed as maximization of a variational lower-bound on average log-likelihoods, an approach that VAEs are based upon. This result makes a connection between two most-popular generative models, namely GANs and VAEs. More importantly, our result constructs an explicit probability model for GANs that can be used to compute a lower-bound on sample likelihoods. Our experimental results on various datasets demonstrate that this likelihood surrogate can be a good approximation of the true likelihood function. Although in this paper we mainly focus on understanding the behavior of the sample likelihood surrogate in different datasets, the proposed statistical framework of GANs can be used in various statistical inference applications. For example, our proposed likelihood surrogate can be used as a quantitative measure to evaluate the performance of different GAN architectures, it can be used to quantify the domain shifts, it can be used to select a proper generator class by balancing the bias term vs. variance, it can be used to detect outlier samples, it can be used in statistical tests such as hypothesis testing, etc. We leave exploring these directions for future work.

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

## APPENDIX A  PROOF OF THEOREM 1

Using the Baye's rule, one can compute the log-likelihood of an observed sample $\mathbf{y}$ as follows:

$$\log f_Y(\mathbf{y}) = \log f_{Y|X=\mathbf{x}}(\mathbf{y}) + \log f_X(\mathbf{x}) - \log f_{X|Y=\mathbf{y}}(\mathbf{x}) \tag{A.1}$$

$$= \log C - \ell(\mathbf{y}, G(\mathbf{x})) - \log \sqrt{2\pi} - \frac{\|\mathbf{x}\|^2}{2} - \log f_{X|Y=\mathbf{y}}(\mathbf{x}),$$

where the second step follows from equation 2.4.

Consider a joint density function $\mathbb{P}_{X,Y}$ such that its marginal distributions match $\mathbb{P}_X$ and $\mathbb{P}_Y$. Note that the equation A.1 is true for every $\mathbf{x}$. Thus, we can take the expectation of both sides with respect to a distribution $\mathbb{P}_{X|Y=\mathbf{y}}$. This leads to the following equation:

$$\log f_Y(\mathbf{y}) = \mathbb{E}_{\mathbb{P}_{X|Y=\mathbf{y}}}\left[-\ell(\mathbf{y}, \mathbf{G}(\mathbf{x}))/\lambda + \log C - \frac{1}{2}\log 2\pi - \frac{\|\mathbf{x}\|^2}{2} - \log f_{X|Y=\mathbf{y}}(\mathbf{x})\right] \quad \text{(A.2)}$$

$$= \mathbb{E}_{\mathbb{P}_{X|Y=\mathbf{y}}}\left[-\ell(\mathbf{y}, \mathbf{G}(\mathbf{x}))/\lambda + \log C - \frac{1}{2}\log 2\pi - \frac{\|\mathbf{x}\|^2}{2} - \log f_{X|Y=\mathbf{y}}(\mathbf{x})\right.$$

$$\left. + \log\left(\mathbb{P}_{X|Y=\mathbf{y}}(\mathbf{x})\right) - \log\left(\mathbb{P}_{X|Y=\mathbf{y}}(\mathbf{x})\right)\right]$$

$$= -\mathbb{E}_{\mathbb{P}_{X|Y=\mathbf{y}}}\left[\ell(\mathbf{y}, \mathbf{G}(\mathbf{x}))/\lambda\right] - \frac{1}{2}\log 2\pi + \log C + \mathbb{E}_{\mathbb{P}_{X|Y=\mathbf{y}}}\left[-\frac{\|\mathbf{x}\|^2}{2}\right]$$

$$+ \text{KL}\left(\mathbb{P}_{X|Y=\mathbf{y}}\|f_{X|Y=\mathbf{y}}\right) + H\left(\mathbb{P}_{X|Y=\mathbf{y}}\right),$$

where $H(.)$ is the Shannon-entropy function.

Next we take the expectation of both sides with respect to $\mathbb{P}_Y$:

$$\mathbb{E}\left[\log f_Y(Y)\right] = -\frac{1}{\lambda}\mathbb{E}_{\mathbb{P}_{X,Y}}\left[\ell(\mathbf{y}, G(\mathbf{x}))\right] - \frac{1}{2}\log 2\pi + \log C + \mathbb{E}_{f_X}\left[-\frac{\|\mathbf{x}\|^2}{2}\right] \quad \text{(A.3)}$$

$$+ \mathbb{E}_{\mathbb{P}_Y}\left[\text{KL}\left(\mathbb{P}_{X|Y=\mathbf{y}}\|f_{X|Y=\mathbf{y}}\right)\right] + H\left(\mathbb{P}_{X,Y}\right) - H\left(\mathbb{P}_Y\right).$$

Here, we replaced the expectation over $\mathbb{P}_X$ with the expectation over $f_X$ since one can generate an arbitrarily large number of samples from the generator. Since the KL divergence is always non-negative, we have

$$\mathbb{E}\left[\log f_Y(Y)\right] \geq -\frac{1}{\lambda}\left\{\mathbb{E}_{\mathbb{P}_{X,Y}}\left[\ell(\mathbf{y}, \mathbf{G}(\mathbf{x}))\right] - \lambda H\left(\mathbb{P}_{X,Y}\right)\right\} + \log C - \log(m) - \frac{r + \log 2\pi}{2} \quad \text{(A.4)}$$

Moreover, using the data processing inequality, we have $H(\mathbb{P}_{X,Y}) \geq H(\mathbb{P}_{\mathbf{G}(X),Y})$ (Cover & Thomas, 2012). Thus,

$$\underbrace{\mathbb{E}\left[\log f_Y(Y)\right]}_{\text{sample likelihood}} \geq -\frac{1}{\lambda}\underbrace{\left\{\mathbb{E}_{\mathbb{P}_{X,Y}}\left[\ell(\mathbf{y}, \mathbf{G}(\mathbf{x}))\right] - \lambda H\left(\mathbb{P}_{Y,\hat{Y}}\right)\right\}}_{\text{GAN objective with entropy regularizer}} + \log C - \log(m) - \frac{r + \log 2\pi}{2} \quad \text{(A.5)}$$

This inequality is true for every $\mathbb{P}_{X,Y}$ satisfying the marginal conditions. Thus, similar to VAEs, we can pick $\mathbb{P}_{X,Y}$ to maximize the lower bound on average sample log-likelihoods. This leads to the entropic GAN optimization 2.3.

## APPENDIX B  TIGHTNESS OF THE BOUND

In Theorem 1, we showed that the Entropic GAN objective maximizes a lower-bound on the average sample log-likelihoods. This result is in the same flavor of variational lower bounds used in VAEs, thus providing a connection between these two areas. One drawback of VAEs in general is about the lack of tightness analysis of the employed variational lower bounds.

In this section, we aim to understand the tightness of the entropic GAN lower bound for some generative models. Corollary 2 shows that the entropic GAN lower bound is tight when $\text{KL}\left(\mathbb{P}_{X|Y=\mathbf{y}}\|f_{X|Y=\mathbf{y}}\right)$ approaches 0. Quantifying this term can be useful for assessing the quality of the proposed likelihood surrogate function. We refer to this term as the approximation gap.

Computing the approximation gap can be difficult in general as it requires evaluating $f_{X|Y=\mathbf{y}}$. Here we perform an experiment for linear generative models and a quadratic loss function (same setting of Corrolary 1). Let the real data $Y$ be generated from the following underlying model

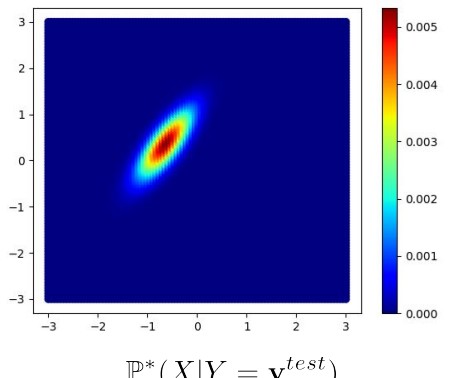 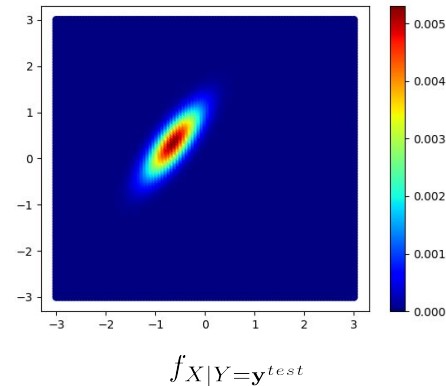

$$\mathbb{P}^*(X|Y = \mathbf{y}^{test}) \qquad\qquad f_{X|Y=\mathbf{y}^{test}}$$

Figure 4: A visualization of density functions of $\mathbb{P}_{X|Y=\mathbf{y}^{\text{test}}}$ and $f_{X|Y=\mathbf{y}^{\text{test}}}$ for a random two-dimensional $\mathbf{y}^{\text{test}}$. Both distributions are very similar to one another making the approximation gap (i.e. $\text{KL}\left(\mathbb{P}_{X|Y=\mathbf{y}^{test}} \| f_{X|Y=\mathbf{y}^{test}}\right)$) very small. Our other experimental results presented in Table 1 are consistent with this result.

$$f_{Y|X=\mathbf{x}} \sim \mathcal{N}(\mathbf{Gx}, \lambda\mathbf{I})$$
$$\text{where } X \sim \mathcal{N}(0, \mathbf{I})$$

Using the Bayes rule, we have

$$f_{X|\mathbf{y}^{test}} \sim \mathcal{N}(\mathbf{Ry}^{test}, \mathbf{I} - \mathbf{RG})$$
$$\text{where } \mathbf{R} = \mathbf{G}^T(\mathbf{GG}^T + \lambda\mathbf{I})^{-1}$$

Since we have a closed-form for $f_{X|Y}$, $\text{KL}\left(\mathbb{P}_{X|Y=\mathbf{y}} \| f_{X|Y=\mathbf{y}}\right)$ can be computed efficiently.

The matrix $\mathbf{G}$ to generate $Y$ is chosen randomly. Then, an entropic GAN with a linear generator and non-linear discriminators are trained on this dataset. $\mathbb{P}_{X|Y=\mathbf{y}}$ is then computed using equation 4.3. Table 1 reports the average surrogate log-likelihood values and the average approximation gaps computed over 100 samples drawn from the underlying data distribution. We observe that the approximation gap is orders of magnitudes smaller than the log-likelihood values.

Additionally, in Figure 4, we demonstrate the density functions of $\mathbb{P}_{X|Y=\mathbf{y}}$ and $f_{X|Y=\mathbf{y}}$ for a random $\mathbf{y}$ and a two-dimensional case ($r = 2$). In this figure, one can observe that both distributions are very similar to one another making the approximation gap very small.

Architecture and hyper-parameter details: For the generator network, we used 3 linear layers without any non-linearities ($2 \to 128 \to 128 \to 2$). Thus, it is an over-parameterized linear system. The discriminator architecture (both $D_1$ and $D_2$) is a 2-layer MLP with ReLU non-linearities ($2 \to 128 \to 128 \to 1$). $\lambda = 0.1$ was used in all the experiments. Both generator and discriminator were trained using the Adam optimizer with a learning rate $10^{-6}$ and momentum 0.5. The discriminators were trained for 10 steps per generator iteration. Batch size of $512$ was used.

Table 1: The tightness of the entropic GAN lower bound. Approximation gaps are orders of magnitudes smaller than the surrogate log-likelihood values. Results are averaged over 100 samples drawn from the underlying data distribution.

| Noise Dimension | Approximation Gap | Surrogate Log-Likelihood |
|:---:|:---:|:---:|
| 2 | $9.3 \times 10^{-4}$ | $-4.15$ |
| 5 | $4.7 \times 10^{-2}$ | $-15.35$ |
| 10 | $6.2 \times 10^{-2}$ | $-46.3$ |

---

**Algorithm 1** Estimating sample likelihoods in GANs

---

1: Sample $N$ points $\mathbf{x}_i \overset{i.i.d}{\sim} P_X(\mathbf{x})$
2: Compute $u_i := \mathbb{P}_X(\mathbf{x}_i) \exp\left(v^*\left(\mathbf{y}^{\text{test}}, G^*(\mathbf{x}_i)\right)/\lambda\right)$
3: Normalize to get probabilities $p_i = \frac{u_i}{\sum_{i=1}^N u_i}$

4: Compute $L = -\frac{1}{\lambda}\left[\sum_{i=1}^N p_i l(\mathbf{y}^{\text{test}}, G^*(\mathbf{x}_i)) + \lambda \sum_{i=1}^N p_i \log p_i\right] - \sum_{i=1}^N p_i \frac{\|\mathbf{x}_i\|^2}{2}$
5: Return $L$

---

## APPENDIX C    OPTIMAL COUPLING FOR W2GAN

Optimal coupling $\mathbb{P}^*_{Y,\hat{Y}}$ for the W2GAN (quadratic GAN (Feizi et al., 2017)) can be computed using the gradient of the optimal discriminator (Villani, 2008) as follows.

**Lemma 2** *Let $\mathbb{P}_Y$ be absolutely continuous whose support contained in a convex set in $\mathbb{R}^d$. Let $\mathbf{D}^{opt}$ be the optimal discriminator for a given generator $\mathbf{G}$ in W2GAN. This solution is unique. Moreover, we have*

$$\hat{Y} \overset{dist}{=} Y - \nabla\mathbf{D}^{opt}(Y), \tag{C.1}$$

*where $\overset{dist}{=}$ means matching distributions.*

## APPENDIX D    SINKHORN LOSS

In practice, it has been observed that a slightly modified version of the entropic GAN demonstrates improved computational properties (Genevay et al., 2017; Sanjabi et al., 2018). We explain this modification in this section. Let

$$W_{\ell,\lambda}(\mathbb{P}_Y, \mathbb{P}_{\hat{Y}}) := \min_{\mathbb{P}_{Y,\hat{Y}}} \mathbb{E}\left[\ell(Y, \hat{Y})\right] + \lambda D_{\text{KL}}\left(\mathbb{P}_{Y,\hat{Y}}\right), \tag{D.1}$$

where $D_{\text{KL}}(.\|.)$ is the KullbackLeibler divergence. Note that the objective of this optimization differs from that of the entropic GAN optimization 2.3 by a constant term $\lambda H(\mathbb{P}_Y) + \lambda H(\mathbb{P}_{\hat{Y}})$. A sinkhorn distance function is then defined as (Genevay et al., 2017):

$$\bar{W}_{\ell,\lambda}(\mathbb{P}_Y, \mathbb{P}_{\hat{Y}}) := 2W_{\ell,\lambda}(\mathbb{P}_Y, \mathbb{P}_{\hat{Y}}) - W_{\ell,\lambda}(\mathbb{P}_Y, \mathbb{P}_Y) - W_{\ell,\lambda}(\mathbb{P}_{\hat{Y}}, \mathbb{P}_{\hat{Y}}). \tag{D.2}$$

$\bar{W}$ is called the Sinkhorn loss function. Reference Genevay et al. (2017) has shown that as $\lambda \to 0$, $\bar{W}_{\ell,\lambda}(\mathbb{P}_Y, \mathbb{P}_{\hat{Y}})$ approaches $W_{\ell,\lambda}(\mathbb{P}_Y, \mathbb{P}_{\hat{Y}})$. For a general $\lambda$, we have the following upper and lower bounds:

**Lemma 3** *For a given $\lambda > 0$, we have*

$$\bar{W}_{\ell,\lambda}(\mathbb{P}_Y, \mathbb{P}_{\hat{Y}}) \le 2W_{\ell,\lambda}(\mathbb{P}_Y, \mathbb{P}_{\hat{Y}}) \le \bar{W}_{\ell,\lambda}(\mathbb{P}_Y, \mathbb{P}_{\hat{Y}}) + \lambda H(\mathbb{P}_Y) + \lambda H(\mathbb{P}_{\hat{Y}}). \tag{D.3}$$

**Proof**  From the definition equation D.2, we have $W_{\ell,\lambda}(\mathbb{P}_Y, \mathbb{P}_{\hat{Y}}) :\ge \bar{W}_{\ell,\lambda}(\mathbb{P}_Y, \mathbb{P}_{\hat{Y}})/2$. Moreover, since $W_{\ell,\lambda}(\mathbb{P}_Y, \mathbb{P}_Y) \le H(\mathbb{P}_Y)$ (this can be seen by using an identity coupling as a feasible solution for optimization D.1) and similarly $W_{\ell,\lambda}(\mathbb{P}_{\hat{Y}}, \mathbb{P}_{\hat{Y}}) \le H(\mathbb{P}_{\hat{Y}})$, we have $W_{\ell,\lambda}(\mathbb{P}_Y, \mathbb{P}_{\hat{Y}}) \le \bar{W}_{\ell,\lambda}(\mathbb{P}_Y, \mathbb{P}_{\hat{Y}})/2 + \lambda/2 H(\mathbb{P}_Y) + \lambda/2 H(\mathbb{P}_{\hat{Y}})$. ∎

Since $H(\mathbb{P}_Y) + H(\mathbb{P}_{\hat{Y}})$ is constant in our setup, optimizing the GAN with the Sinkhorn loss is equivalent to optimizing the entropic GAN. So, our likelihood estimation framework can be used with models trained using Sinkhorn loss as well. This is particularly important from a practical standpoint as training models with Sinkhorn loss tends to be more stable in practice.

## APPENDIX E    TRAINING ENTROPIC GANS

In this section, we discuss how WGANs with entropic regularization is trained. As discussed in Section 3, the dual of the entropic GAN formulation can be written as

$$\min_{\mathbf{G} \epsilon \mathcal{G}} \max_{\mathbf{D}_1, \mathbf{D}_2} \mathbb{E}\left[\mathbf{D}_1(Y)\right] - \mathbb{E}\left[\mathbf{D}_2(\mathbf{G}(X))\right] - \lambda \mathbb{E}_{\mathbb{P}_Y \times \mathbb{P}_{\hat{Y}}} \left[\exp\left(v(\mathbf{y}, \hat{\mathbf{y}})/\lambda\right)\right],$$

where

$$v(\mathbf{y}, \hat{\mathbf{y}}) := \mathbf{D}_1(\mathbf{y}) - \mathbf{D}_2(\hat{\mathbf{y}}) - \ell(\mathbf{y}, \hat{\mathbf{y}}).$$

We can optimize this min-max problem using alternating optimization. A better approach would be to take into account the smoothness introduced in the problem due to the entropic regularizer, and solve the generator problem to stationarity using first-order methods. Please refer to Sanjabi et al. (2018) for more details. In all our experiments, we use Algorithm 1 of Sanjabi et al. (2018) to train our GAN model.

### E.1   GAN's Training on MNIST

MNIST dataset constains $28 \times 28$ grayscale images. As a pre-processing step, all images were resized in the range $[0, 1]$. The Discriminator and the Generator architectures used in our experiments are given in Tables. 2,3. Note that the dual formulation of GANs employ two discriminators - $D_1$ and $D_2$, and we use the same architecture for both. The hyperparameter details are given in Table 4. Some sample generations are shown in Fig. 5

### E.2   GAN's Training on CIFAR

We trained a DCGAN model on CIFAR dataset using the discriminator and generator architecture used in Radford et al. (2015). The hyperparamer details are mentioned in Table. 5. Some sample generations are provided in Figure 7

Table 2: Generator architecture

| Layer | Output size | Filters |
|---|---|---|
| Input | 128 | - |
| Fully connected | 4.4.256 | $128 \to 256$ |
| Reshape | $256 \times 4 \times 4$ | - |
| BatchNorm+ReLU | $256 \times 4 \times 4$ | - |
| Deconv2d ($5 \times 5$, str 2) | $128 \times 8 \times 8$ | $256 \to 128$ |
| BatchNorm+ReLU | $128 \times 8 \times 8$ | - |
| Remove border row and col. | $128 \times 7 \times 7$ | - |
| Deconv2d ($5 \times 5$, str 2) | $64 \times 14 \times 14$ | $128 \to 64$ |
| BatchNorm+ReLU | $128 \times 8 \times 8$ | - |
| Deconv2d ($5 \times 5$, str 2) | $1 \times 28 \times 28$ | $64 \to 1$ |
| Sigmoid | $1 \times 28 \times 28$ | - |

Table 3: Discriminator architecture

| Layer | Output size | Filters |
|---|---|---|
| Input | $1 \times 28 \times 28$ | - |
| Conv2D($5 \times 5$, str 2) | $32 \times 14 \times 14$ | $1 \to 32$ |
| LeakyReLU(0.2) | $32 \times 14 \times 14$ | - |
| Conv2D($5 \times 5$, str 2) | $64 \times 7 \times 7$ | $32 \to 64$ |
| LeakyReLU(0.2) | $64 \times 7 \times 7$ | - |
| Conv2d ($5 \times 5$, str 2) | $128 \times 4 \times 4$ | $64 \to 128$ |
| LeakyRelU(0.2) | $128 \times 4 \times 4$ | - |
| Reshape | 128.4.4 | - |
| Fully connected | 1 | $2048 \to 1$ |

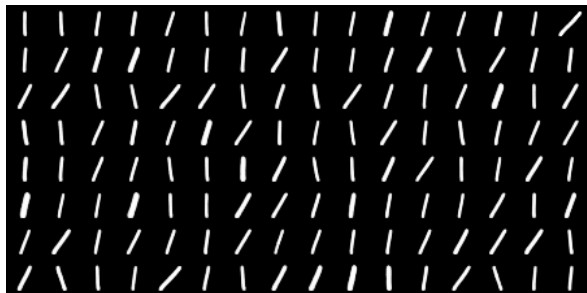

Figure 5: Samples generated by Entropic GAN trained on MNIST

Figure 6: Samples generated by Entropic GAN trained on MNIST-1 dataset

### E.3  GAN's Training on LSUN-Bedrooms dataset

We trained a WGAN model on LSUN-Bedrooms dataset with DCGAN architectures for generator and discriminator networks (Arjovsky et al., 2017). The hyperparameter details are given in Table. 6, and some sample generations are provided in Fig. 8

Table 4: Hyper-parameter details for MNIST experiment

| Parameter | Config |
|---|---|
| $\lambda$ | 5 |
| Generator learning rate | 0.0002 |
| Discriminator learning rate | 0.0002 |
| Batch size | 100 |
| Optimizer | Adam |
| Optimizer params | $\beta_1 = 0.5, \beta_2 = 0.9$ |
| Number of critic iters / gen iter | 5 |
| Number of training iterations | 10000 |

Table 5: Hyper-parameter details for CIFAR-10 experiment

| Parameter | Config |
|---|---|
| Generator learning rate | 0.0002 |
| Discriminator learning rate | 0.0002 |
| Batch size | 64 |
| Optimizer | Adam |
| Optimizer params | $\beta_1 = 0.5, \beta_2 = 0.99$ |
| Number of training epochs | 100 |

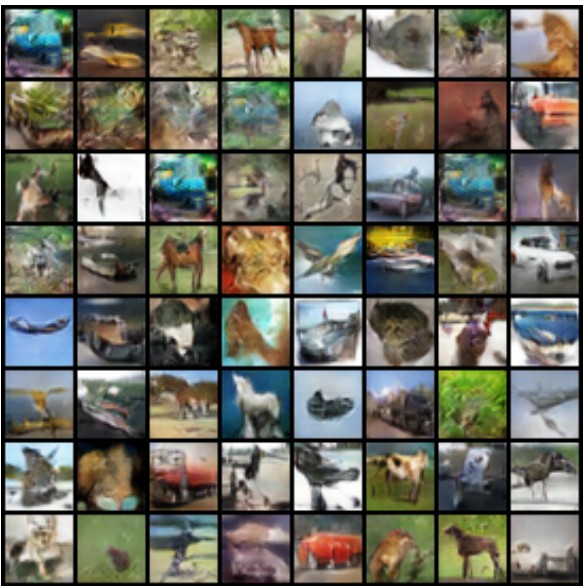

Figure 7: Samples generated by DCGAN model trained on CIFAR dataset

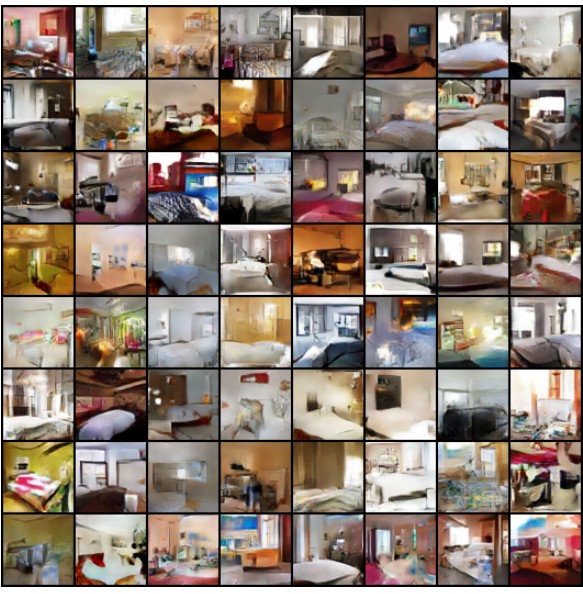

Figure 8: Samples generated by WGAN model trained on LSUN-Bedrooms dataset

Table 6: Hyper-parameter details for LSUN-Bedrooms experiment

| Parameter | Config |
|---|---|
| Generator learning rate | 0.00005 |
| Discriminator learning rate | 0.00005 |
| Clipping parameter $c$ | 0.01 |
| Number of critic iters per gen iter | 5 |
| Batch size | 64 |
| Optimizer | RMSProp |
| Number of training iterations | 70000 |

