# OpenReview forum: "Entropic GANs meet VAEs: A Statistical Approach to Compute Sample Likelihoods in GANs"
_ICLR.cc/2019/Conference_

### Official Review · AnonReviewer2 · 2018-11-01
**Interesting attempt on theory of entropic GANs**

**Rating:** 6
**Confidence:** 5

**Review:**

The contribution of the paper is to show that WGAN with entropic regularization maximize a lower bound on the likelihood of the observed data distribution. While the WGAN formulation minimizes the Wasserstein distance of the transformed latent distribution and the empirical distribution which is already a nice measure of "progress", having a bound on the likelihood can be interesting.

Pros:
+ I like the entropic GAN formulation and believe it is very interesting as it gives access to the joint distribution of latent and observed variables.
+ While there are some doubtful statements, overall the paper is well written and easy to read.

Cons:
- The assumption of injectivity of the generator could be problematic, as it might not be fulfilled due to mode collapse.
- I feel the theory is not very deep. Since one has a closed form of the transportation map (Eq. 3.7), the likelihood of the data is obtained by marginalizing out the latent space. However, this assumes that the inner dual maximization problem is solved to stationarity so that Eq 3.7 holds, which is not the case in practice (5 discriminator updates).
- Thus in Sec. 4.1 for the likelihood at various points in training it is not clear what is actually happening.
- Sec 4.3 for unregularized GANs might be problematic. In general, the transportation plan is not a density function, so I'm not certain whether Theorem 1 / Corollary 2 still hold. Furthermore, the heuristic for "inverting" G^* is very crude.

- There are also some minor problematic statements in the paper. While they can be easily fixed, they give me doubts:
  * The original VAE paper is not cited in the introduction for VAEs
  * The 2013 paper by Cuturi cited on page 2 has nothing to do with "computational aspects of GANs". It is about fast computation of approximate OT between two discrete prob. measures.
  * First-order / second-order Wasserstein distance is I think a bit unusual name for W_1, W_2
  * On pg. 4, the point of the entropy term is to make the objective strongly convex. Strict convexity has no computational benefits.

---

> ### Author Response · Authors · 2018-11-13
> **Authors' response**
>
> We thank the reviewer for the valuable comments. We have addressed them in the revised version of the paper. Below, we provide point to point responses to the comments:
>
> Pros:
> Thank you for these comments.
>
> Cons:
> (1) Assumption of injectivity: Thank you for this comment. We agree that having an injective G was a strong assumption. Fortunately, we do not need this assumption at all. In the revised version of the paper, we show that our results hold without this assumption. In order to do this, we use the data-processing inequality from Information Theory which indicates that for a random variable X, the entropy of G(X) is always less or equal to the entropy of X. For more details, please see equation A.5 and Section 4 in the revised paper.
>
> (2) Stationarity of inner dual problem: Thanks for the comments. First, the likelihood surrogate is computed using Corollary 2 which has three terms visualized in Fig. 1. Marginalizing the transportation map of Eq. 3.7 is a necessary step to compute the likelihood surrogate (this part corresponds to the encoder part in VAE formulations).
>
> We compute the surrogate likelihoods only after the generative model is trained. In our approach, Entropic GANs are trained using Algorithm 1 of (Sanjabi et al, NIPS 2018) (as mentioned in Appendix E). It has been shown in Theorem 4.2 of that paper that Algorithm 1 leads to a close approximation of stationary solutions of the Entropic GAN objective. We have added further explanations about this to the revised paper.
>
> (3) Likelihood computation at intermediate iterations: We thank the reviewer for pointing out that the stationarity assumption will not be satisfied at the intermediate iterations of training, and hence likelihood computations may not be accurate. To fix this, we re-ran the discriminator updates for 100 steps before computing the surrogate likelihoods at intermediate iterations. We obtained almost the same behavior in likelihoods. We have updated plots in the revised version of the paper and have added further explanations about this experiment.
>
> (4)	Unregularized GANs: Thank you for the comment. First note that the coupling P_X|y (i.e. the transportation map) is always a valid density function. Second note that our results in Theorem 1 and Corollary 2 hold for a general GAN formulation (not just the entropic GAN). However, for a general GAN, it may not be easy to compute P_X|y using GAN’s dual formulation. For the entropic GAN, eq 3.7 gives a closed form relationship between GAN’s dual solutions and P_X|y. For un-regularized GANs, in some special cases that we discuss in Appendix C, such a closed-form relationship exists (but not in general). In experiments of Section 4.3, we ‘approximate’ P_X|y with a delta function spiked on the closest latent sample generating y. The heuristic used in this section imitates eq 3.7 by taking into account both the likelihood of the latent variable as well as the distance between y and the model. In general, understanding the behavior of the optimal coupling P_X|y using GAN’s dual solutions is an interesting direction for the future work.
>
> (5) Minor comments: Thank you for pointing out these typos. We have modified the paper accordingly. The names used for W_1 and W_2 (the first and second order Wasserstein distances) are common names used in the optimal transport literature. For example, see Villani’s book titled “Optimal transport: old and new”. However, we agree that in the machine learning literature, these names are less common. For example, WGAN (Wasserstein GAN) is in fact using the first-order Wasserstein distance in its formulation. We have added further explanations about these names to the revised version of the paper.

---

### Official Review · AnonReviewer1 · 2018-11-01
**Interesting idea, but need more polishing**

**Rating:** 5
**Confidence:** 4

**Review:**


1. The assumption made by the authors that "generator is injective" is problematic or even wrong, as it is well known that GAN suffers from mode collapsing problem.

2. It is very confusing when the authors mentioned the negative Shannon entropy. Because the equation the authors wrote is the Shannon entropy, not the negative version.

3. In the 5th paragraph in the  introduction section, the paper (Cuturi, 2013) has nothing to do with "improve computational aspect of GAN", maybe the authors want to cite this paper "Learning Generative Models with Sinkhorn Divergences".

4. The authors failed to discuss their paper with "ON THE QUANTITATIVE ANALYSIS OF DECODERBASED GENERATIVE MODELS", which uses AIS to estimate the likelihood.

Suggestion:
1. Please use \cdot instead of , i.e. F(\cdot) instead of F(.)
2. Typo: in Appendix ?? and ??, in section 4

---

> ### Author Response · Authors · 2018-11-13
> **Authors' response**
>
> We thank the reviewer for the valuable comments. We have addressed them in the revised version of the paper. Below, we provide point to point responses to the comments:
>
> (1)	Thank you for this comment. We agree that having an injective G was a strong assumption. Fortunately, we do not need this assumption at all. In the revised version of the paper, we show that our results hold without this assumption. In order to do this, we use the data-processing inequality from Information Theory which indicates that for a random variable X, the entropy of G(X) is always less or equal to the entropy of X. For more details, please see equation A.5 and Section 4 in the revised paper.
>
> (2)	You are right about the definition of the Shannon entropy. What we meant in that phrase was that the strongly-convex regularization term is the negative Shannon entropy, i.e. -H(P_{Y,\hY}) (because the Shannon entropy is a concave function and we need convexity in minimization). We have clarified this in the revised version of the paper.
>
> (3)	Thank you for pointing out this typo. We have updated the paper accordingly.
>
> (4)	Thank you for pointing out this relevant reference. We have added a discussion about it to the introduction of the revised version of the paper.

---

### Official Review · AnonReviewer3 · 2018-11-05
**Interesting connection, but lacks clarity**

**Rating:** 5
**Confidence:** 3

**Review:**

Summary
The authors notice that entropy regularized optimal transport produce an upper bound of a certain model likelihood. Then, the authors claim it is possible to leverage that upper bound to come up with a measure of 'sample likelihood', the probability of a certain sample under the model.

Evaluation
The idea is certainly interesting and novel, as it allows to bridge two distinct worlds (VAE and GANs). However, I am concerned about the message (or lack of thereof) that is conveyed in the paper. Particularly, the following two points makes me be reluctant to recommend an acceptance:

1)There is no measure on the tightness of the lower bound. How can we tell if this bound isnt tight? All results are dependent on the bound being close to the true value. No comments about this are given.
2)The sample likelihoods are dependent on a certain "model". Here the nomenclature is confusing because I thought GANS were a probabilistic model, but now there is an additional model regarding a function f. How these two relate? What happens if I change f? to which extent the results depend on f?
3)related to 2): the histograms in figure 2 are interesting, but they are not conclusive that the measure that is being proposed is a 'bona fide' sample likelihood.

---

> ### Author Response · Authors · 2018-11-13
> **Authors' response**
>
> We thank the reviewer for the valuable comments. We have addressed them in the revised version of the paper. Below, we provide point to point responses to the comments:
>
> (1)	Tightness of the entropic GAN lower bound: in Theorem 1, we showed that the Entropic GAN objective provides a lower-bound on the average sample log-likelihoods. This result is in the same flavor of variational lower bounds used in VAEs, thus providing a principled connection between GANs and VAEs. One drawback of VAEs (which has been echoed in reviewer’s comment 1) is about the lack of the tightness analysis of the employed variational lower bound.
> To address this comment, we have empirically analyzed the tightness of the entropic GAN lower bound for some simple generative models. We have explained our results in detail in a new Appendix Section (Appendix B) which includes a table (Table 1) and a figure (Figure 4). Briefly, our result in Corollary 2 indicates that the approximation gap can be quantified as the KL divergence between P_{X|Y=\by} (the latent variable distribution resulted from the entropic GAN optimization) and f_{X|Y=\by} (the latent variable distribution according to the true model of the data). We evaluate this approximation gap for a linear generative model and a quadratic loss function. Our empirical results show that the approximation gap is orders of magnitudes smaller than the log-likelihood values (see Table 1 and Figure 4). This approach can potentially be used in the tightness analysis of VAEs as well.
>
> (2)	Thank you for your comment. Let us explain our result and the data model a bit further. A classical approach to compute a generative model using some observed samples is to consider a parametric family of density functions (referred to as f(.), the data model) and optimize its parameters using maximum likelihood. VAEs are approximations of this approach. GANs, however, seemingly do not take this traditional approach to the generative problem. GANs compute a generative distribution $G^*(X)$ that minimizes a distance (such as the optimal transport distance) to the observed distribution. However, GANs do not make any density assignments to the points outside of the range of G^* and that is the key issue because after training a GAN, if we observe a new point y^{test}, it is very likely that this point does not lie exactly on the range of G^*. Thus, GANs are unable to assign a reasonable probability to this point. Intuitively, we can imagine that if this point is ‘close’ to G^*(X), it is more likely to be generated from this model. Our result provides a theoretically justified way to define what we mean by being ‘close’ to G^*.
>
> Our key idea is to consider an explicit model for the data in GAN’s framework so that we can compute sample likelihoods. Similar to VAEs, f(.) in our model is the underlying data distribution. We assume that the data is generated as per Eq. 2.4 using a ground truth (and unknown) function G. This is a reasonable model for the data since G can be a complex function. By training the entropic GAN, we essentially estimate this function using the generator network of GANs. We have added further explanations about this to the introduction of the paper.
>
> (3)	The histograms are plotted using the likelihood estimator presented in Corollary 2. As mentioned in Corollary 2, the proposed estimator of sample likelihood approaches true likelihoods when the KL divergence term approaches 0. The newly included Appendix B section presents empirical evidence that the approximation error is orders of magnitude smaller than the likelihood values for linear models. Also, in the real image datasets where computing true likelihoods are difficult, our proposed estimator exhibits sensible trends indicating that the proposed estimator is a good estimator of sample likelihoods.

---

### Author Response · Authors · 2018-11-26
**Summary of the authors' response**

We thank the reviewers for their comments. In summary, there were two key comments raised by the reviewers that we have addressed as follows (details have been posted in our point by point responses to the comments):

(1) The assumption that “the generator function is injective” has been relaxed. By using the data processing inequality from information theory, we show that our results hold even without this assumption.
(2) We have added a new section with empirical analysis highlighting the tightness of the entropic GAN lower bound for the log-likelihood function.

---

### Meta-Review · Area_Chair1 · 2018-12-18

**Confidence:** 4
**Recommendation:** Reject

**Metareview:**

The paper's strength is in that it shows the log likelihood objective is lower bounded by a GAN objective plus an entropy term. The theory is novel (but it seems to relate closely to the work https://arxiv.org/abs/1711.02771.) The main drawback the reviewer raised includes a) it's not clear how tight the lower bound is; b) the theory only applies to a particular subcase of GANs --- it seems that the only reasonable instance that allows efficient generator is the case where Y = G(x)+\xi where \xi is Gaussian noise. The authors addressed the issue a) with some new experiments with linear generators and quadratic loss, but it lacks experiments with deep models which seems to be necessary since this is a critical issue. Based on this, the AC decided to recommend reject and would encourage the authors to add more experiments on the tightness of the lower bound with bigger models and submit to other top venues.